# Hierarchical Ni-Mn LDHs@CuC_2_O_4_ Nanosheet Arrays-Modified Copper Mesh: A Dual-Functional Material for Enhancing Oil/Water Separation and Supercapacitors

**DOI:** 10.3390/ijms241814085

**Published:** 2023-09-14

**Authors:** Yue Wu, Guangyuan Lu, Ping Xu, Tian C. Zhang, Huaqiang He, Shaojun Yuan

**Affiliations:** 1Low-Carbon Technology & Chemical Reaction Engineering Lab, College of Chemical Engineering, Sichuan University, Chengdu 610065, Chinaxp001108@163.com (P.X.); hehuaqiang_scu@163.com (H.H.); 2Civil & Environmental Engineering Department, University of Nebraska-Lincoln, Omaha, NE 68182-0178, USA; tzhang1@unl.edu

**Keywords:** superwetting mesh, self-cleaning, oil/water separation, supercapacitors

## Abstract

The pursuit of superhydrophilic materials with hierarchical structures has garnered significant attention across diverse application domains. In this study, we have successfully crafted Ni-Mn LDHs@CuC_2_O_4_ nanosheet arrays on a copper mesh (CM) through a synergistic process involving chemical oxidation and hydrothermal deposition. Initially, CuC_2_O_4_ nanosheets were synthesized on the copper mesh, closely followed by the growth of Ni-Mn LDHs nanosheets, culminating in the establishment of a multi-tiered surface architecture with exceptional superhydrophilicity and remarkable underwater superoleophobicity. The resultant Ni-Mn LDHs@CuC_2_O_4_ CM membrane showcased an unparalleled amalgamation of traits, including superhydrophilicity, underwater superoleophobicity, and the ability to harness photocatalytic forces for self-cleaning actions, making it an advanced oil-water separation membrane. The membrane’s performance was impressive, manifesting in a remarkable water flux range (70 kL·m^−2·^h^−1^) and an efficient oil separation capability for both oil/water mixture and surfactant-stabilized emulsions (below 60 ppm). Moreover, the innate superhydrophilic characteristics of the membrane rendered it a prime candidate for deployment as a supercapacitor cathode material. Evidenced by a capacitance of 5080 mF·cm^−2^ at a current density of 6 mA cm^−2^ in a 6 M KOH electrolyte, the membrane’s potential extended beyond oil-water separation. This work not only introduces a cutting-edge oil-water separation membrane and supercapacitor electrode but also offers a promising blueprint for the deliberate engineering of hierarchical structure arrays to cater to a spectrum of related applications.

## 1. Introduction

The rapid advancement of industrialization worldwide has led to detrimental consequences for the ecological environment, such as the discharge of oily wastewater and greenhouse gas emissions [1,2]. Consequently, efficient water treatment and energy storage systems have gained increasing significance. Superwetting membranes, created by controlling the chemical composition and microstructure of the membrane surface, have become a popular approach for treating oily wastewater due to their cost-effectiveness and high flow flux [3]. The deliberate design of a vertically arranged nanosheet structure can maintain water stability and form a stable three-phase interface of oil/water/solid, resulting in excellent oil rejection performance [4,5,6]. Moreover, the microstructure design is crucial for constructing ion diffusion pathways in aqueous alkali-based energy storage devices [6,7]. The construction of electrodes with a hierarchical structure has been demonstrated to facilitate the reduction of ion diffusion pathways and mitigate the volume effect during charge/discharge processes [7,8]. This presents a promising opportunity to rationally design a hierarchical structured material capable of both oil-water separation and energy storage applications.

Layered double hydroxides (LDHs) are a family of layered structure materials composed of interlayer balancing anions and positively charged lamellar cations such as Al^3+^, Mn^3+^, and Fe^3+^ [9]. These cations partially replace bivalent metal cations such as Ni^2+^, Co^2+^, and Fe^2+^, which are coordinated octahedrally by hydroxyl groups [10]. Anions such as Cl^-^, NO_3_^−^, SO_4_^2−^, CO_3_^2−^, and RCO^2−^ balance the positive charge of the cations [10,11,12]. Combining LDHs with high porosity membrane substrates is an attractive strategy for constructing functionalized membranes with hierarchical structures [13,14,15]. The good catalytic reduction performance of LDHs endows the membrane with excellent antifouling and self-cleaning properties, enabling it to remove surfactant-stabilized emulsions and dyes from the residue on the membrane [16,17,18]. Yin et al. [15] reported the successful fabrication of a Ni-Co LDH on the surface of a stainless-steel mesh, displaying a robust self-cleaning oil-repellent ability without hydration. Sun et al. [19] proposed a dual-functional mesh with Zn-Ni-Co LDHs@NiMoO_4_ heterojunction nanoarrays, which could efficiently separate various oil/water mixtures with high flux and also exhibited good photocatalytic performance for the degradation of organic dyes. The large interlayer spacing of LDHs makes it favorable for the diffusion of ions and water molecules, thereby promoting electrochemical reaction kinetics in aqueous supercapacitors (SCs) [20]. Ruan et al. [21] prepared Ni(OH)_2_/Cu_2_O/CuO nanoclusters on a nickel foam, which had a capacitance of 1474 F g^−1^ at 15 mA cm^−2^ and could retain a capacitance of 82% after 1500 cycles. Zhang et al. [22] reported that core-shell structured NiMn-LDHs@CuO on copper foam delivered a capacitance of 2430 F g^−1^ at 0.8 A g^−1^.

Accordingly, we hypothesized that a substrate with a nanosheet array structure could serve as an advanced functionalized material with efficient oil-water separation and good supercapacitor performance. The aim of this study was to fabricate a novel dual-functional membrane for simultaneously enhancing oil/water separation and supercapacitors. As illustrated schematically in Figure 1, the growth of CuC_2_O_4_ nanosheet arrays was accomplished on the surface of a copper mesh (CuC_2_O_4_ CM) by a chemical etching method. Subsequently, Ni-Mn LDHs were deposited on the CuC_2_O_4_ CM by a hydrothermal method, forming a hierarchically structured Ni-Mn LDHs@CuC_2_O_4_ CM. The introduction of Ni-Mn LDHs increased the surface roughness of the membrane and improved its emulsion-breaking performance. The Ni-Mn LDHs@CuC_2_O_4_ CM showed a photo-catalytically driven self-cleaning function for degrading oil pollution while maintaining its superhydrophilicity. This material showed an ultrahigh separation flux (up to 7.0 × 10^4^ L m^−2^ h^−1^) with residual oil contents in filtrates below 60 mg·L^-1^ for oil/water mixtures and a flux of 2000 L·m^−2^·h^−1^ with that below 100 mg L^−1^ for surfactant-stabilized oil-in-water emulsions. As a cathode for SCs, the LDHs@CuC_2_O_4_ CM delivered a capacitance of 5080 mF cm^−2^ at a current density of 6 mA cm^−2^ and still exhibited a capacitance retention of 80.25% after 800 cycles at a current density of 15 mA cm^−2^. Our study unveils a pioneering oil-water separation membrane and supercapacitor electrode, embodying the forefront of innovation. Beyond this, we present a compelling blueprint for the intentional design of hierarchical structure arrays poised to serve an array of interconnected applications.

## 2. Results and Discussion

### 2.1. Characterization of the Ni-Mn LDHs@CuC_2_O_4_ CM

Figure 1 schematically illustrates the two-step synthesis process to obtain a hierarchical Ni-Mn LDHs@CuC_2_O_4_ CM sample. First, CuC_2_O_4_ nanosheet arrays are fabricated by chemical etching, providing a larger specific surface area and more sites to grow Ni-Mn LDHs on the mesh framework. Then, the CuC_2_O_4_ nanosheet array-coated copper mesh is transported to a hydrothermal reactor for Ni-Mn LDHs growth. The CuC_2_O_4_ nanosheets function as the core for in-situ growth of Ni-Mn LDHs@ CuC_2_O_4_, and all as-prepared products maintain a sheets-like array throughout the reaction process. 

The representative SEM images of the as-prepared Ni-Mn LDHs@CuC_2_O_4_ CM at different hydrothermal times are shown in Figure 2. Compared with Appendix A, it is clearly observed that Ni-Mn LDHs nanosheets grow in situ on the CuC_2_O_4_ nanosheets and cover the CuC_2_O_4_ CM substrates completely after 8 h of hydrothermal reaction (Figure 2d). As the hydrothermal process proceeded for a specific time (i.e., 2, 4, 6, and 8 h) (Figure 2a–d), these Ni-Mn LDHs continuously grew to give rise to dense nanosheets with a height and a width not exceeding 10 μm. At a reaction time of 2 h, a delicate layer of Ni-Mn LDHs nanosheets forms on the CuC_2_O_4_ CM substrates (Figure 2a), signifying the successful construction of hierarchical nanostructures. With progressive increments in reaction time, a noticeable proliferation of Ni-Mn LDHs nanosheets becomes apparent on the mesh membrane, culminating in a more compact and comprehensive hierarchical arrangement (Figure 2b,c). Remarkably, at an 8-h reaction time, distinct particles comprised of Ni-Mn LDHs nanosheets densely populate the mesh membrane substrates (Figure 2d), thereby amplifying superhydrophilicity through increased membrane surface area and heightened surface roughness (Cassie Model). Furthermore, these nanosheets grow vertically, forming a grid-like staggered structure, which provides abundant reaction sites for reversible Faraday redox reactions and a solid backbone for charge/electrolyte ion transport [23,24]. At the same time, due to the low height (about 1–2 μm) and the interconnected structure of the nanosheet, the contact between the electrode material on the surface and the collector is closer, which is conducive to improving the charge transport and electrochemical performance of the nanosheet.

Figure 3 shows that the Ni-Mn LDHs@CuC_2_O_4_ CM after 8 h hydrothermal reaction has 22.2%, 3.3%, 22.1%, 47.8%, and 4.5% for Ni, Mn, C, O, and Cu, respectively, indicating that Ni-Mn LDHs is uniformly loaded on CuC_2_O_4_ CM in the hydrothermal reaction. Additionally, the contents of Ni and Mn elements are higher than the contents of reaction time 6 h shown in Appendix A, confirming that the nanosheet structure is denser and more homogeneous as the hydrothermal reaction time increases.

Figure 4a shows that the XRD pattern of oxalic acid-treated copper mesh has the characteristic peaks of CuC_2_O_4_ (PDF: 21-0297), demonstrating the successful synthesis of CuC_2_O_4_ micro-nano sheets. After the hydrothermal reaction, new diffraction peaks located at 11.3° were observed, which should be attributed to the (0 0 3) planes of Ni-Mn LDHs. The surface chemical composition of Ni-Mn LDHs@CuC_2_O_4_ CM can be further determined by XPS. The peaks of Cu, Ni, Mn, O, and C shown in the wide scan spectrum indicate the presence of these elements in the Ni-Mn LDHs@CuC_2_O_4_ CM (Figure 4b). The Cu 2p core level spectra in Figure 4c and Cu (0) and Cu (II) with binding energy (i.e., BE) at 933.1 and 935.3 eV, respectively, correspond to Cu substrate and CuC_2_O_4_ nanosheets [25]. The Ni^2+^ 2p3/2 and Ni^2+^ 2p1/2 peaks accompanied bands are located at 855.6 eV and 873.2 eV, respectively, which indicate the presence of Ni^2+^ in Ni-Mn LDHs@CuC_2_O_4_ CM (Figure 4d) [22,23]. In addition, the Mn 2p1/2 and Mn 2p3/2 are located at 641.9 and 653.1 eV, respectively, suggesting that the main oxidation state of the Mn cation in the Ni-Mn LDHs is Mn^3+^ ions (Figure 4e) [22,23]. In the O 1s spectrum, the dominating peak with BE at 531.1 eV is associated with the -OH, and the other small component peak with BE at 532.4 eV corresponds to the oxalate ions C_2_O_4_^2−^ (Figure 4f) [25]. In C 1s core level spectra (Appendix A), the peak component with BE at 284.7 eV is assigned as adventitious carbon and is used as a correction for the other peak components. The peak component with the BE at 286.3 eV is attributed to the C-O bond, consistent with the characteristic bonding of the oxalate ion during the transformation process. The peak with the BE at 289.1 eV corresponds to the carboxyl group in C_2_O_4_^2−^ [26].

### 2.2. Surface Wettability Characterization

The selective wettability of the novel mesh membrane plays a vital role in efficient oil/water separation [26,27]. A hierarchical micro-nano rough structure with high surface energy was constructed on the mesh surface. Hence, a superhydrophilic/ underwater superhydrophobic surface was successfully formed. The wettability of the membrane was tested by the measurement of static (or dynamic) WCAs. Figure 5a shows that a water droplet can quickly spreading-wetting within 80 ms with a static WCA at 0° on the Ni-Mn LDHs@CuC_2_O_4_ CM surface. By contrast, the static WCA of pure CM was 90°, as shown in Appendix A. Similarly, Figure 5b shows that the underwater oil contact angles (UWOCAs) of all kinds of oil species are greater than 150°. Among them, the underwater OCAs of 1,2-dichloroethane, kerosene, cyclohexane, and isooctane are 157 ± 3°, 156 ± 4°, 158 ± 1°, and 155 ± 1°, respectively, indicating the superior underwater superoleophobicity of the Ni-Mn LDHs@CuC_2_O_4_ CM for general applicability. Additionally, an ultralow underwater oil sliding angle (UWOSA) (8°) can be realized on the surface of Ni-Mn LDHs@CuC_2_O_4_ CM (Figure 6a), which may imply an extreme underwater oil anti-adhesion property of the mesh. Thus, the oil anti-adhesion test was conducted. When red dichloroethane oil drops and kerosene oil drops were dropped onto the membrane under underwater conditions, the oil droplets bounced back quickly and fell off without adhering to the membrane (Figure 6b,c), which resulted from the low adhesion of oil droplets on the underwater superoleophobic surface of the Ni-Mn LDHs@CuC_2_O_4_ CM. All these phenomena directly confirm the excellent underwater oil anti-adhesion property of Ni-Mn LDHs@CuC_2_O_4_ CM.

### 2.3. Oil-Water Separation Performance

The specific separation performance was tested by a series of confirmatory experiments (Figure 7). The separation experiments for oil/water mixtures were conducted by a vertical tubular device (Figure 7a). The feed column height was always maintained at about 15.5 cm, which is equivalent to a static pressure of about 1.5 kPa. The membrane was pre-wetted by water to ensure the formation of a stable water film on the membrane surface to isolate oil. The separation result is shown in Figure 7a,b. When the oil/water mixture is added from above the tube, due to the limited contact between the oil and the membrane surface, the oil is confined in the upper separation tube, while the water phase passes rapidly through the membrane and is collected in the beaker below the separating tube. Figure 7b shows that the collected water is clear and transparent, and almost all of the oil phase is retained by the membrane and stays in the upper separation tube (Figure 7a). The CTAB-stabilized oil-in-water emulsions (SSEs) prepared by different oil types were used for the separation, and the results are shown in Figure 7c–f. As expected, the numerous sub-micrometer and micrometer oil droplets in all milky feeds are completely removed (Figure 7d–f), indicating that Ni-Mn LDHs@CuC_2_O_4_ CM can separate the SSEs efficiently.

To further determine the separation performance, the water permeation flux and the COD value in the filtrate were measured. Figure 8a shows that the permeate fluxes of the cyclohexane, kerosene, and isooctane are 98.22, 76.39, and 85.94 × 10^3^ L m^−2^ h^−1^, respectively. Residual oil concentrations (expressed as COD values) in the filtrate of the cyclohexane, kerosene, and isooctane are as low as approximately 51.15, 12.03, and 7.52 mg L^−1^, respectively. Furthermore, the difference in the flux and the COD value in filtrate can be observed among the SSEs derived from different oil types due to the different physicochemical properties of different oil (Figure 8b). Specifically, the permeate fluxes of cyclohexane, kerosene, and isooctane emulsions are 2291.83, 1909.86, and 1762.95 L m^−2^ h^−1^, respectively. The corresponding COD value in the filtrate is 96.29, 91.77, and 76.73 mg L^−1^, respectively. Compared to the separation of oil/water mixtures, the permeate fluxes for the separation of the surfactant-stabilized emulsion decrease sharply, and the residual oil concentrations in the filtrate increase greatly. The reason for this is probably that the pores of the mesh membrane are blocked by the filter cake consisting of difficult-to-gather oil droplets. To comprehensively evaluate the oil/water separation capabilities of our Ni-Mn LDHs@CuC_2_O_4_ CM, we present a succinct performance analysis encompassing permeation flux and separation efficiency, as compared to previously documented separation membranes (Table 1) [27,28,29,30]. Evidently, the Ni-Mn LDHs@CuC_2_O_4_ CM exhibits a comparable efficiency in separating oil/water mixtures while significantly outpacing numerous reported membranes in terms of water permeation flux. This underscores the exceptional oil-water separation proficiency of the Ni-Mn LDHs@CuC_2_O_4_ CM, a feat further magnified by the strategic integration of hierarchical Ni-Mn LDH structures.

### 2.4. Photocatalytic Self-Cleaning and Stability

The photocatalytic self-cleaning capability of the separation membrane surface is particularly important for a separation system with long-term oil/water separation [32]. The photocatalysis self-cleaning performance of the Ni-Mn LDHs@CuC_2_O_4_ CM is shown in Figure 8c,d. The membrane was immersed in kerosene to construct a membrane surface contaminated with oil, and the contaminated membrane has lost its superhydrophilic surface properties with a WCA of 133.8°. After 80 min of UV light illumination, the superhydrophilicity of the fouling membrane can be completely restored, demonstrating the good photocatalytic self-cleaning performance of the membrane.

In practical applications, the stability of the membrane is very important for the separation system. Based on water and salt resistance, this study explores the long-term useability of the novel membrane in freshwater and saltwater environments. The membrane was placed in deionized water, and the underwater oil contact angle was tested every day. Furthermore, the membrane was immersed in 1% to 5% NaCl solution to test the salt tolerance. Comfortingly, whether in a water environment (Figure 8e) or a high-salt environment (Figure 8f), the meshes always maintained relatively stable underwater superoleophobicity (all UWOCAs > 150°), indicating a favorable chemical tolerance of Ni-Mn LDHs@CuC_2_O_4_ CM.

### 2.5. Electrochemical Performance

Figure 9a shows a pair of peaks in CV curves within the potential range of −0.3–0.7 V, indicating the battery-like charge storage mechanism. This is consistence with the previous reports [33,34]. With an increase in scanning rate, a faster redox rate occurs on the electrode materials, gradually increasing the redox peak current. The anode peak shifts slightly to the high potential region, while the cathode peak shifts slightly to the low potential region, implying that electrochemical polarization occurs as the scan rate increases. GCD measurements were carried out on the Ni-Mn LDHs@CuC_2_O_4_ CM at different current densities. Figure 9b shows that the shape of the GCD curve is an approximately symmetrical triangle, implying good columbic efficiency and faradaic pseudo capacitance performance [35,36]. Based on the GCD data, the area-specific capacitance of Ni-Mn LDHs@CuC_2_O_4_ CM at different current densities is illustrated in Figure 9c. Clearly, a specific capacitance of 5080, 4711, 4376, and 3753 mF cm^−2^ can be achieved at current densities of 6, 8, 10, and 15 mA cm^−2^. Figure 9d shows the Nyquist plot of the Ni-Mn LDHs@CuC_2_O_4_ CM electrode, and the result is fitted by an equivalent electrical circuit (Figure 9d inset). The Nyquist plot consists of a small semicircle in the high-frequency region and a straight line in the low-frequency region, reflecting the transfer of changes at the electrode/electrolyte interface and the ion diffusion process in the electrode, respectively [37,38]. The internal resistance (R_s_) and the charge transfer resistance (R_ct_) were calculated to be 0.7 and 1.7 Ω, respectively, indicating the good electron transfer stability of the Ni-Mn LDHs@CuC_2_O_4_ CM electrode. The long cycling test was also carried out, and the specific capacitance of each cycle was collected based on the GCD data. Notably, Figure 9e shows that at a current density of 15 mA cm^−2^, a capacitance retention of 80.25% was achieved after 800 cycles, indicating the good electrochemical stability of the Ni-Mn LDHs@CuC_2_O_4_ CM electrode. For comparison, Table 2 outlines the capacitive performance of the as-prepared Ni-Mn LDHs@CuC_2_O_4_ CM electrodes alongside other electrodes documented in the existing literature [39,40,41,42,43]. The data clearly illustrate that the Ni-Mn LDHs@CuC_2_O_4_ electrode holds its own against the capacitive benchmarks set by recently reported supercapacitors. This outcome underscores the exceptional energy storage capabilities of our Ni-Mn LDHs@CuC_2_O_4_ CM electrode, affirming that the adept assembly of hydrotalcite significantly enhances its energy storage prowess.

## 3. Materials and Methods

### 3.1. Materials

The commercial copper mesh (>99.9% purity, 500 meshes) was purchased from Shengzhuo Wire Mesh Co. (Hebei, China). Chemical reagents of reagent grade, such as oxalic acid (H_2_C_2_O_4_.2H_2_O), manganese chloride (MnCl_2_.4H_2_O), nickel chloride (NiCl_2_.6H_2_O), hexamethylenetetramine, acetone, isopropanol, ethanol, hydrochloric acid (36 wt%), and hexadecyl trimethyl ammonium bromide (CTAB), were procured from Kelong Chemical Co. (Chengdu, China). Oil samples, including kerosene, isooctane, 1,2-dichloroethane, and cyclohexane, were obtained from Aladdin Reagent Co. (Shanghai, China). Deionized water (18.25 MΩ·cm) was generated using a commercial reverse osmosis (RO) workstation.

### 3.2. Preparation of CuC_2_O_4_ Nanosheet Arrays

The well-defined CuC_2_O_4_ nanosheet arrays were prepared by a previously reported chemical etching method [25]. The copper mesh was firstly pretreated to remove grease and oxide layers by immersing the copper mesh into a solution (containing each acetone, isopropyl alcohol, anhydrous ethanol, and deionized water) followed by ultrasonic treatment (energy level = 40 kHz, 240 W) for 10 min each. The mesh was then placed in 1 M HCl for 15 min to remove the oxide layer on the mesh surface. Finally, the mesh was placed in 1 M H_2_C_2_O_4_ solution for chemical etching for 7 days at 70 °C. Afterward, it was rinsed with a copious amount of deionized water and dried in a vacuum oven at 60 °C overnight.

### 3.3. Preparation of Ni-Mn LDHs@CuC_2_O_4_ CM

The growth of Ni-Mn LDHs on the surfaces of CuC_2_O_4_ CM was achieved via a well-established hydrothermal reaction [44]. Typically, 3 mmol of NiCl_2_, 1 mmol of MnCl_2,_ and 5 mmol of hexamethylenetetramine were dissolved in 30 mL of deionized water and stirred evenly for 30 min. The solution was then transferred into a 50 mL Teflon-lined stainless-steel reactor with the prepared CuC_2_O_4_ CM and subjected to a hydrothermal reaction at 90 °C for (predetermined) 2, 4, 6, or 8 h. After the reaction, the collected Ni-Mn LDHs@ CuC_2_O_4_ CM was washed with deionized water and dried in a vacuum oven at 60 °C for 24 h.

### 3.4. Characterization

Scanning electron microscopy (SEM, Regulus 8230, Hitachi, Japan), a matched energy dispersive spectrometer (EDS), and X-ray diffraction (XRD, DX2700, Haoyuan Instruments Co., China) were used to assess the surface morphology, chemical composition, and crystalline structure of Ni-Mn LDHs@CuC_2_O_4_ CM, respectively. The XRD analysis was conducted using Cu Kα radiation with a wavelength of 1.5418 Å. The surface chemical compositions and valence states were characterized by X-ray photoelectron spectroscopy (XPS) with an Axis Ultra Has XPS spectrometer (Kratos Analytical Co., Wharfside, Manchester, UK). A contact angle measuring instrument (JC2000, Shanghai Zhongchen Digital Equipment Co., Shanghai, China) was used, with a 3 µL droplet of liquid applied to the mesh membrane surface, to obtain the contact angle values by averaging the results from three measurements at different positions. 

### 3.5. Determination of Oil/Water Separation Performance

The test experiment to assess the oil and water separation performance of Ni-Mn LDHs@CuC_2_O_4_ CM was conducted in four steps. First, the material was fixed in a tubular oil/water separation device for oil/water mixtures or a pump oil/water separation device for emulsions. Second, the material was pre-moistened with deionized water. Third, different samples were poured from the upper end of the device into the (pump) device, from which water samples were collected. Finally, the chemical oxygen demand (COD) of those samples was measured using a COD spectrophotometer. The gravity-driven separation performance of the as-prepared Ni-Mn LDHs@CuC_2_O_4_ CM was evaluated on various oil-water mixtures and oil-in-water emulsions. 30 mL of cyclohexane dyed with Sudan Ⅲ was mixed with 30 mL of deionized water, and 2 mL of kerosene was mixed with 198 mL of deionized water to create an oil-water mixture and an oil-in-water emulsion, respectively. To stabilize the emulsion, an aliquot of 300 mg of CTAB was added to the mixture and then stirred for 2 h. The emulsion separation flux was calculated using the following Equation:F = V/(S × t)(1)
where V is the filtrate volume per unit time (kL), S is the area of the flange port of the separation device (m^2^), and t is the separation time (h).

### 3.6. Photocatalytic-Driven Self-Cleaning Ability and Stability Testing

To measure the photocatalytic-driven self-cleaning ability of Ni-Mn LDHs@CuC_2_O_4_ CM under visible light irradiation, the as-prepared mesh membrane was immersed in the kerosene solution for a while (10 min) to obtain an oil-contaminated copper mesh, which was then placed on a glass sheet and transferred to a photocatalytic reactor for photodegradation using a 500 W ZJB 380 xenon lamp with a 400 nm cut-off filter as the visible light source. The photocatalytic degradation ability was determined by measuring the change in water contact angles (WCAs) as a function of irradiation time. The chemical stability of the resulting Ni-Mn LDHs@CuC_2_O_4_ CM was evaluated by monitoring its change in the underwater oil contact angle over time when it was immersed in a 1%, 2%, 3%, 4% or 5% NaCl solution and deionized water for 6 and 7 days, respectively. 

### 3.7. Electrochemical Energy Storage Performance Testing

The electrochemical energy storage performance of the Ni-Mn LDHs@CuC_2_O_4_ CM was investigated using cyclic voltammetry (CV), galvanostatic charge-discharge (GCD), and electrochemical impedance spectroscopy (EIS). The Ni-Mn LDHs@CuC_2_O_4_ CM sample with a dimension of 1 × 1 cm was used as the working electrode, the Ag/AgCl electrode as the reference electrode, and the Pt electrode as the counter electrode, with a 6 M KOH solution as the electrolyte. The EIS measurement was performed at open-circuit voltage and an AC voltage amplitude of 5 mV in a frequency range of 0.01–100 kHz on a CHI-660E electrochemical workstation. On the basis of the GCD curves, the specific capacitance of the sample was calculated by the following Equation:C = (I × ∆t)/(S × ∆V)(2)
where I is the discharging current (mA), Δt is the discharging time (s), ΔV denotes the voltage range, and S corresponds to the area of active material in the working electrode (cm^2^). The three parameters I, Δt, and ΔV were obtained from GCD curves.

## 4. Conclusions

This study presented a novel approach to fabricating Ni-Mn LDHs@CuC_2_O_4_ nanosheet arrays on a copper mesh, achieved through a combination of chemical oxidation and hydrothermal deposition. The resulting structure served dual purposes as an oil-water separation membrane and a supercapacitor (SC) cathode. The Ni-Mn LDHs@CuC_2_O_4_ CM, acting as an advanced oil-water separation membrane, exhibited an impressive array of properties, including superhydrophilicity, underwater superoleophobicity, and photocatalytic self-cleaning capabilities. Remarkably, the membrane demonstrated a separation flux for oil/water mixtures reaching up to 70 kL m^−2^ h^−1^ while effectively reducing residual oil contents in the filtrate to below 60 mg L^−1^. When faced with surfactant-stabilized oil-in-water emulsions, the Ni-Mn LDHs@CuC_2_O_4_ CM maintained a separation flux of approximately 2 kL m^−2^ h^−1^, ensuring residual oil content remains below 100 mg L^−1^. Moreover, the Ni-Mn LDHs@CuC_2_O_4_ CM showcased robust chemical stability during extensive testing. When employed as an SC cathode, it achieved a remarkable capacitance of 5080 mF cm^−2^ at a current density of 6 mA cm^-2^. Impressively, even at a current density of 15 mA cm^−2^, noteworthy capacitance retention of 80.25% was maintained after 800 cycles. This work not only introduces a groundbreaking oil-water separation membrane and SC cathode but also proposes an innovative strategy for designing hierarchical structure arrays, thereby enriching the landscape of relevant applications.

## Figures and Tables

**Figure 1 ijms-24-14085-f001:**
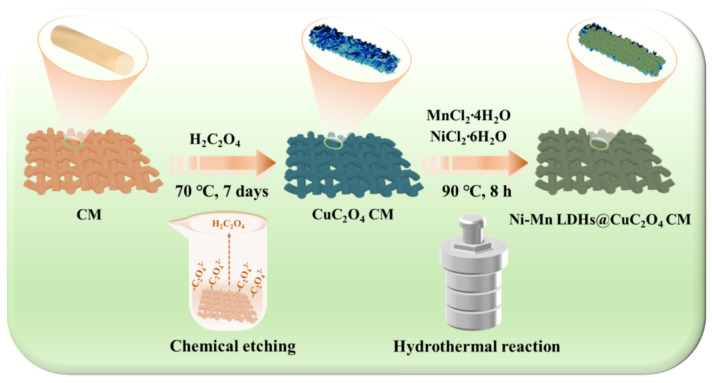
Schematic diagram of a two-step process for preparing Ni-Mn LDHs@CuC_2_O_4_ CM by chemical oxidation and hydrothermal deposition.

**Figure 2 ijms-24-14085-f002:**
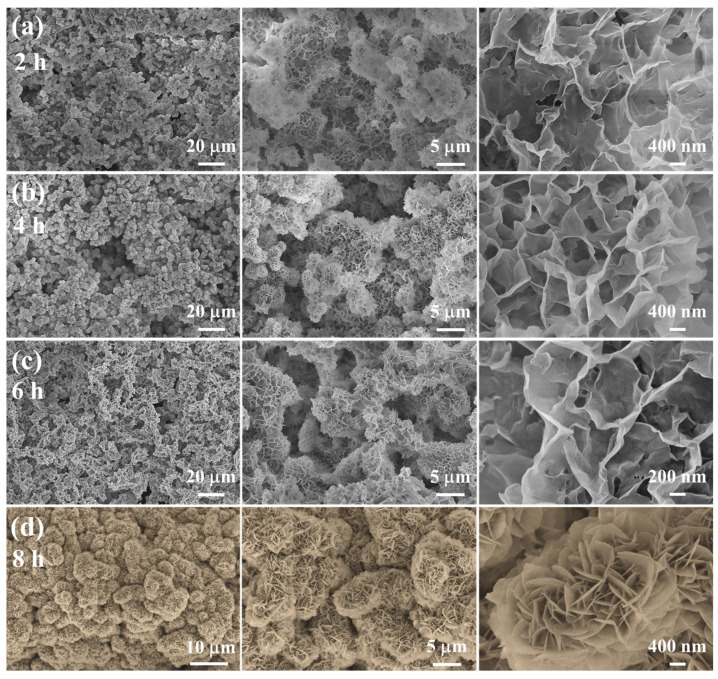
(**a**–**d**) SEM images of different magnifications of Ni-Mn LDHs@CuC_2_O_4_ CM after hydrothermal reaction for 2, 4, 6, or 8 h.

**Figure 3 ijms-24-14085-f003:**
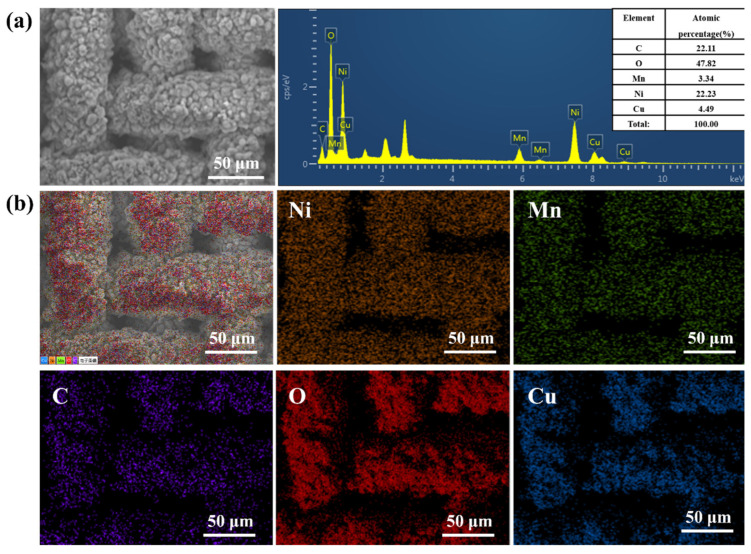
(**a**) The SEM image and corresponding element content, as well as (**b**) EDS mapping of Ni-Mn LDHs@CuC_2_O_4_ CM after 8 h hydrothermal reaction.

**Figure 4 ijms-24-14085-f004:**
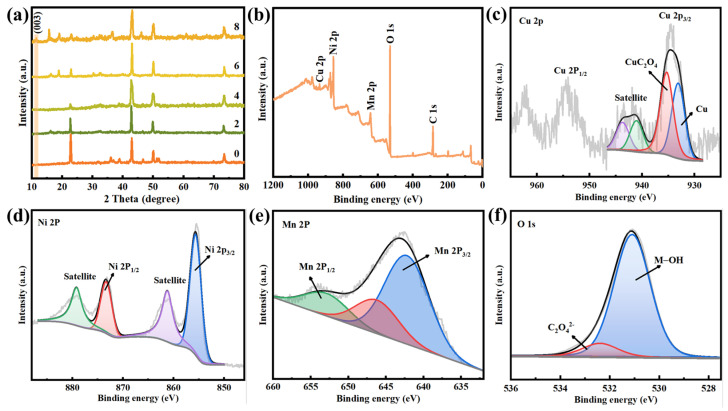
(**a**) XRD patterns of copper oxalate mesh after chemical oxidation reaction and Ni-Mn LDHs@CuC_2_O_4_ CM after hydrothermal reaction for 2, 4, 6, or 8 h. XPS spectrums of (**b**) wide scan, (**c**) Cu 2p, (**d**) Ni 2p, (**e**) Mn 2p, and (**f**) O 1s of Ni-Mn LDHs@CuC_2_O_4_ CM after 8-h hydrothermal reaction.

**Figure 5 ijms-24-14085-f005:**
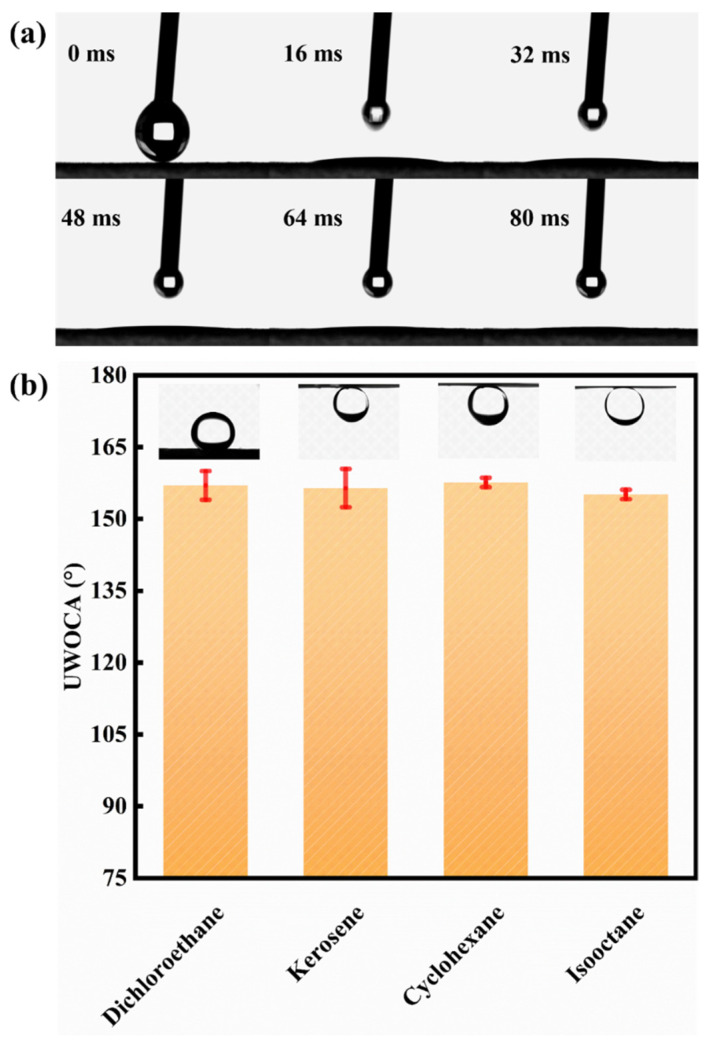
(**a**) Dynamic WCA snapshot of water droplets in the air on the surface of Ni-Mn LDHs@CuC_2_O_4_ CM. (**b**) Underwater oil contact angle of Ni-Mn LDHs@CuC_2_O_4_ CM for different oil types.

**Figure 6 ijms-24-14085-f006:**
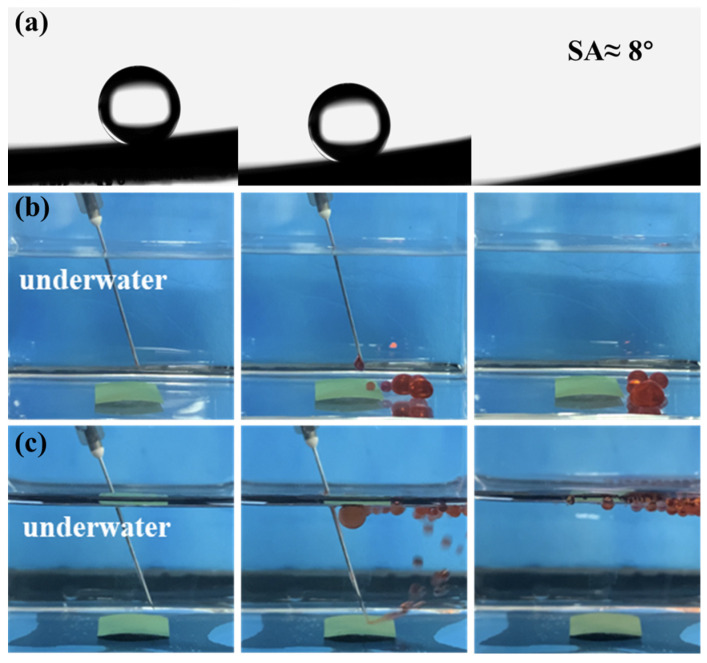
(**a**) Underwater dynamic oil adhesion angle test of Ni-Mn LDHs@CuC_2_O_4_ CM. (**b**,**c**) Underwater oil pollution resistance test, using dichloroethane and kerosene as oil pollution, respectively.

**Figure 7 ijms-24-14085-f007:**
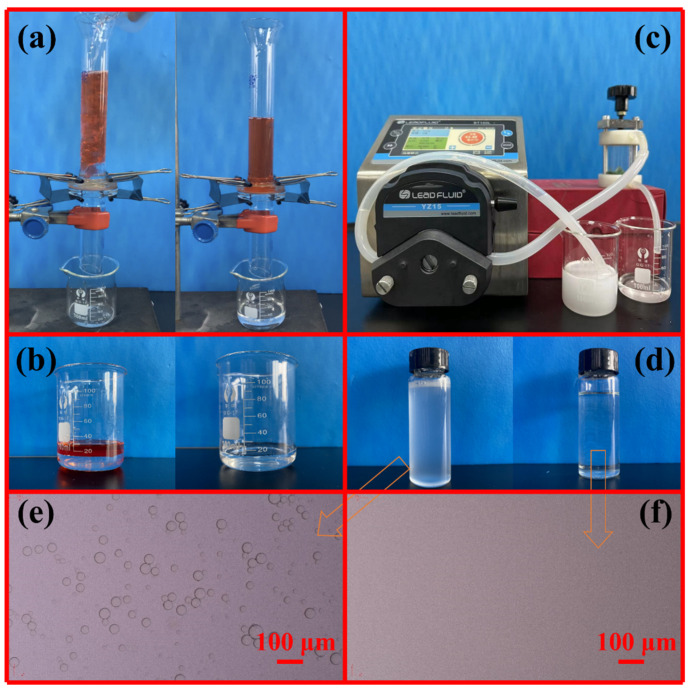
Photographs of the separation devices and results: (**a**) oil-water mixture; (**c**) oil-in-water emulsion; comparison before and after separation of (**b**) oil-water mixture and (**d**) emulsion, and (**e**,**f**) the corresponding optical micrograph of emulsion and filtrate.

**Figure 8 ijms-24-14085-f008:**
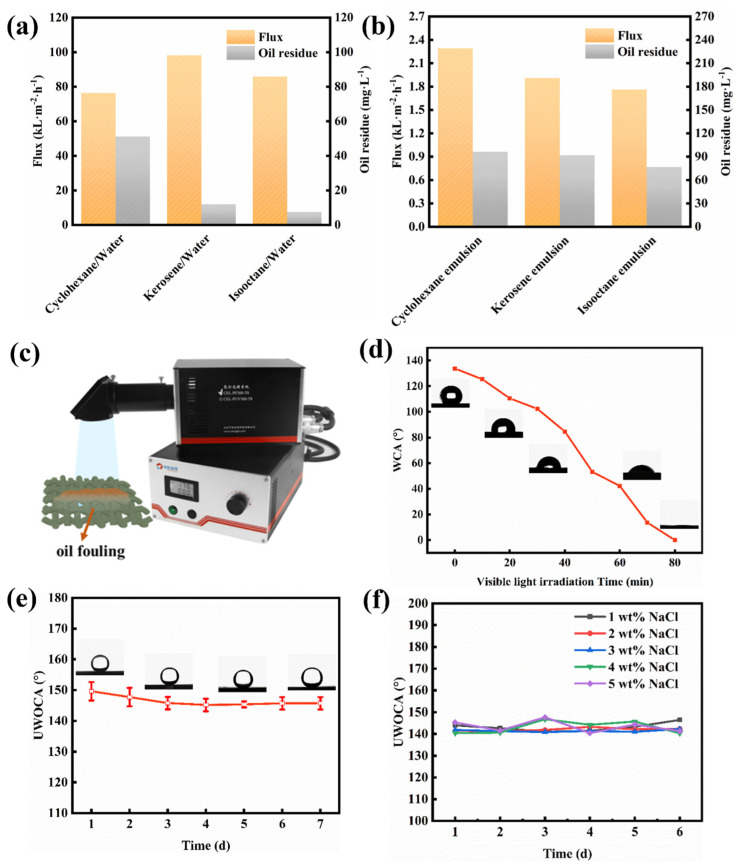
Ni-Mn LDHs@CuC_2_O_4_ CM separation of (**a**) oil-water mixtures and (**b**) surfactant stabilized emulsions: variation in flux and residual oil content (COD value) with different oil types. (**c**) A Xenon lamp emitter was used. (**d**) The effect of visible light irradiation time on WCA after oil-water separation with Ni-Mn LDHs@CuC_2_O_4_ CM. The stability test: underwater oil contact angle changes in Ni-Mn LDHs@CuC_2_O_4_ CM after being immersed into (**e**) deionized water and (**f**) NaCl solution with different concentrations.

**Figure 9 ijms-24-14085-f009:**
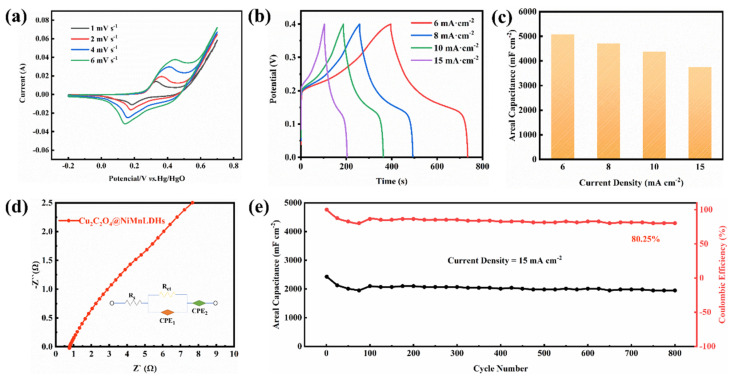
The electrochemical performance of Ni-Mn LDHs@CuC_2_O_4_ CM: (**a**) CV curves at different scan rates; (**b**) GCD curves at different current densities; (**c**) the specific capacitance based on GCD data; (**d**) Nyquist plots; and (**e**) long cycling test at 15 mA cm^−2^ for 800 cycles.

**Table 1 ijms-24-14085-t001:** Performance comparison of Ni-Mn LDHs@CuC_2_O_4_ and various membranes for separating O/W emulsions.

No.	Separation Membrane	Flux (L·m^−2^·h^−1^)	Separation Efficiency	Reference
1	g-C_3_N_4_/Ti(OH)_4_/PFOA	317.2	95%	[27]
2	Ti_3_C_2_T_X_ MXene-PAN	1573	98.6%	[28]
3	CL-LPDA-SiO_2_@PDA-CM	109.76	97%	[29]
4	ZnO/WO_3_.H_2_O	431	96%	[30]
5	Zn-Ni-Co LDHs@NiMoO_4_	1981	>98%	[19]
6	CuC_2_O_4_@ Cu-MOFs	1800	>99.0%	[31]
7	Ni-Mn LDHs@CuC_2_O_4_	2292	>99.0%	This work

**Table 2 ijms-24-14085-t002:** Performance Comparison of Ni-Mn LDHs@CuC_2_O_4_ and various supercapacitors for capacitive performance.

No.	Separation Membrane	Capacitive Performance	Reference
1	NiMn LDH@NiCo_2_O_4_/CC	2.40 F cm^−2^ at 20 mA cm^−2^	[39]
2	CoNi_2_S_4_/CC	3.16 F cm^−2^ at 10 mA cm^−2^	[40]
3	Fe-Co-S/P	5.06 F cm^−2^ at 20 mA cm^−2^	[41]
4	CuMoP	5.2 F cm^−2^ at 3 mA cm^−2^	[42]
5	Ni_7_S_6_/CoNi_2_S_4_@CF	2.84 F cm^−2^ at 20 mA cm^−2^	[43]
6	Ni-Mn LDHs@CuC_2_O_4_	5.08 F cm^−2^ at 6 mA cm^−2^	This work

## Data Availability

The data that support the findings of this study are available from the corresponding author upon reasonable request.

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
