# Peer review of "Hierarchical Ni-Mn LDHs@CuC2O4 Nanosheet Arrays-Modified Copper Mesh: A Dual-Functional Material for Enhancing Oil/Water Separation and Supercapacitors"

_ijms, 2023, doi:10.3390/ijms241814085_

Round 1
Reviewer 1 Report
The paper "Hierarchical Ni-Mn LDHs@CuC2O4 Nanosheet Arrays-Modified Copper Mesh: A Dual-Functional Material for Enhancing Oil/Water Separation and Supercapacitors" by Yue Wu describes the preparation and characterization of a novel hierarchical Ni-Mn LDHs@CuC2O4 nanosheet array on copper mesh. The material exhibits excellent oil/water separation and supercapacitor properties, making it a promising candidate for dual-functional applications.
This work is well-presented, and the results are clear and convincing. The authors have demonstrated that the Ni-Mn LDHs@CuC2O4 CM is a promising material for both oil-water separation and SCs. The separation performance is excellent, and the capacitance is also comparable to other reported materials. The chemical tolerance is also a major advantage. Overall, this work is a significant contribution to the field, and I recommend it for publication in the International Journal of Molecular Sciences (IJMS) with minor comments.
Minor Comments
- The authors could provide more information about the stability of the material under long-term cycling.
- The authors could also compare the performance of the Ni-Mn LDHs@CuC2O4 nanosheet array to other dual-functional materials that have been reported in the literature.
Author Response
Responses to the Reviewers’ Comments
Reviewer:
The paper "Hierarchical Ni-Mn LDHs@CuC2O4 Nanosheet Arrays-Modified Copper Mesh: A Dual-Functional Material for Enhancing Oil/Water Separation and Supercapacitors" by Yue Wu describes the preparation and characterization of a novel hierarchical Ni-Mn LDHs@CuC2O4 nanosheet array on copper mesh. The material exhibits excellent oil/water separation and supercapacitor properties, making it a promising candidate for dual-functional applications.
This work is well-presented, and the results are clear and convincing. The authors have demonstrated that the Ni-Mn LDHs@CuC2O4 CM is a promising material for both oil-water separation and SCs. The separation performance is excellent, and the capacitance is also comparable to other reported materials. The chemical tolerance is also a major advantage. Overall, this work is a significant contribution to the field, and I recommend it for publication in the International Journal of Molecular Sciences (IJMS) with minor comments.
Response:
We appreciate for the reviewer’s pertinent comments on our manuscript. The manuscript has been carefully revised following the reviewer’s suggestion, and point-by-point response has also provided accordingly.
Reviewer:
The authors could provide more information about the stability of the material under long-term cycling.
Response:
We appreciate the reviewer's valuable suggestion. As recommended, Figure 9e presents the long-term cycling performance of the prepared electrode. The robust electrochemical stability of the Ni-Mn LDHs@CuC2O4 CM electrode is evident through an impressive capacitance retention of 80.25% after 800 cycles at a current density of 15 mA cm-2. To further assess the durability, we subjected the Ni-Mn LDHs@CuC2O4 CM to immersion in deionized water and NaCl solution for 6 days. Encouragingly, no substantial alteration in the water contact angle was observed, confirming the inherent stability of the fabricated samples.
Reviewer:
The authors could also compare the performance of the Ni-Mn LDHs@CuC2O4 nanosheet array to other dual-functional materials that have been reported in the literature.
Response:
We appreciate for the reviewer’s good comments. As suggested, we have supplemented the comparison performance of the oil/water separation and supercapacitance of the as-prepared Ni-Mn LDHs@CuC2O4 CM with literature, and the results are summarized up in Table 1and Table 2. As illustrated in Table 1, the as-prepared Ni-Mn LDHs@CuC2O4 CM shows a comparable separation efficiency of oil/water mixture, and delivers a higher water permeation flux than most of reported membranes (please refer to page 8, lines 224-232, and Table 1). For the sake of comparison, Table 2 outlines the capacitive performance of the as-prepared Ni-Mn LDHs@CuC2O4 CM electrodes alongside other electrodes documented in existing literature. The data clearly illustrates that the Ni-Mn LDHs@CuC2O4 electrode holds its own against the capacitive benchmarks set by recently reported supercapacitors. This outcome underscores the exceptional energy storage capabilities of our Ni-Mn LDHs@CuC2O4 CM electrode, affirming that the adept assembly of hydrotalcite significantly enhances its energy storage prowess (please refer to page 10, lines 285-291, and Table 2).
Reviewer 2 Report
Authors have synthesized hierarchical Ni-Mn LDHs@CuC2O4 Nanosheet Arrays Modified Copper Mesh and explored their applications as supercapacitors and in enhancing 3 Oil/Water Separation.
Authors have performed detailed study in applications section. However, their characterization section is not strong. Authors have performed SEM and EDS experiments in figure 2 and 3. Nice change in size and morphology with reaction time is visible in the figures. But, detailed discussion on size and morphology is missing in manuscript. Authors should elaborate the discussion in manuscript.
Additionally, TEM and AFM characterization of the nanosheet arrays are recommended for better understanding of size, morphology and surface.
English language is fine. Minor checking and editing is required.
Author Response
Responses to the Reviewers’ Comments
Reviewer:
Authors have synthesized hierarchical Ni-Mn LDHs@CuC2O4 Nanosheet Arrays Modified Copper Mesh and explored their applications as supercapacitors and in enhancing Oil/Water Separation.
Authors have performed detailed study in applications section. However, their characterization section is not strong. Authors have performed SEM and EDS experiments in figure 2 and 3. Nice change in size and morphology with reaction time is visible in the figures. But, detailed discussion on size and morphology is missing in manuscript. Authors should elaborate the discussion in manuscript.
Response:
We appreciate for the reviewer’s good suggestions on revising our manuscript. The manuscript has been carefully revised following the reviewer’s suggestion accordingly. As suggested, we have elaborated the discussion on size and morphologies of the as-prepared samples as follows: At a reaction time of 2 hours, a delicate layer of Ni-Mn LDHs nanosheets forms on the CuC2O4 CM substrates (Fig. 2a), signifying the successful construction of hierarchical nanostructures. With progressive increments in reaction time, a noticeable proliferation of Ni-Mn LDHs nanosheets becomes apparent on the mesh membrane, culminating in a more compact and comprehensive hierarchical arrangement (Figs. 2b and 2c). Remarkably, at an 8-hour reaction time, distinct particles comprised of Ni-Mn LDHs nanosheets densely populate the mesh membrane substrates (Fig. 2d), thereby amplifying superhydrophilicity through increased membrane surface area and heightened surface roughness (Cassie Model) (please refer to page 3, lines 107-114).
Reviewer:
Additionally, TEM and AFM characterization of the nanosheet arrays are recommended for better understanding of size, morphology and surface.
Response:
We appreciate the reviewer's valuable suggestions. In fact, characterizing AFM images on a copper mesh membrane poses a challenge for us. This difficulty arises due to the 25 μm pore size of the 500-mesh copper mesh and the rough nature of the copper mesh substrate, which renders it unsuitable for AFM characterization. Since the Ni-Mn LDHs nanosheets are grown on the Cu2C2O4 nanosheet arrays-modified copper mesh, our only option is to detach the Ni-Mn LDHs@Cu2C2O4 from the substrates through ultrasonic shaking. However, this process leads to the disruption of the hierarchical structure of Ni-Mn LDHs@Cu2C2O4. Consequently, distinguishing the size and morphology of the Ni-Mn LDHs in the TEM images becomes challenging. Regrettably, we are unable to provide TEM images as well. Unfortunately, the TEM and AFM tests could not be successfully completed. We kindly request the reviewers' understanding in this matter.
Reviewer 3 Report
In the present report, the authors constructed a Ni-Mn LDHs@CuC2O4 nanosheet array on a copper mesh (CM) via a combined process of chemical oxidation and hydrothermal deposition. The resulting Ni-Mn LDHs@CuC2O4 CM membrane exhibited superior super hydrophilic, underwater superoleophobic and photocatalytically driven self-cleaning properties, making it an advanced oil-water separation membrane. This paper provided some valuable information and the content is very significant in this field. However, I recommended a major revision of the article from its present form before it can be published in ijms. Some specific comments are as follows:
1. The abstract and conclusion sections should be a specific and scientific approach.
2. In the introduction section, the authors should expound the research significance of the present work.
3. The authors should explain the novelty of the present report?
4. The authors should provide a clear schematic representation of the formation mechanism.
5. What is the pH of the reaction solution? The pH of the solution normally varies from precursor to precursor. The authors must justify the selection of pH, temperature and time.
6. At what temperature did the authors perform the hydrothermal process?
7. The authors should include line representation in the XRD pattern.
8. There are many typo mistakes in the text, figure captions and some abbreviations are missing. Please revise carefully.
9. The authors should discuss the effect of morphology and surface area on performance.
10. What are the key factors affecting the efficiency?
11. How do these results influence the previously reported results?
12. The authors should provide an EIS spectra of all samples.
13. Authors should provide CV and GCD curves of all samples.
14. All images are very poor resolution. Authors should produce high quality images.
15. In the current state, there are more typographical errors and the language should be improved. Therefore, the authors are advised to recheck the whole manuscript for improving the language and structure carefully.
In the current state, there are more typographical errors and the language should be improved. Therefore, the authors are advised to recheck the whole manuscript for improving the language and structure carefully.
Author Response
Responses to the Reviewers’ Comments
Reviewer:
In the present report, the authors constructed a Ni-Mn LDHs@CuC2O4 nanosheet array on a copper mesh (CM) via a combined process of chemical oxidation and hydrothermal deposition. The resulting Ni-Mn LDHs@CuC2O4 CM membrane exhibited superior super hydrophilic, underwater superoleophobic and photocatalytically driven self-cleaning properties, making it an advanced oil-water separation membrane. This paper provided some valuable information and the content is very significant in this field. However, I recommended a major revision of the article from its present form before it can be published in ijms. Some specific comments are as follows.
Response:
We appreciate for the reviewer’s pertinent comments on our manuscript. The manuscript has been carefully revised following the reviewer’s suggestion, and point-by-point response has also provided accordingly.
Reviewer:
1. The abstract and conclusion sections should be a specific and scientific approach.
Response:
We are grateful for the reviewer’s insightful suggestion. As suggested, we have rephrased the abstract and conclusion sections in order to describe them in a more specific and scientific manner (please refer to page 1, lines 11-28; page 13, lines 382-399).
Reviewer:
2. In the introduction section, the authors should expound the research significance of the present work.
Response:
We appreciate for the reviewer’s insightful suggestion. As suggested, we have expound the research significance of our study as follows: Our study unveils a pioneering oil-water separation membrane and supercapacitor electrode, embodying the forefront of innovation. Beyond this, we present a compelling blueprint for the intentional design of hierarchical structure arrays, poised to serve an array of interconnected applications (please refer to page 2, line 84-87).
Reviewer:
3. The authors should explain the novelty of the present report?
Response:
We sincerely appreciate the reviewer's thoughtful reminder. Our work entails the creation of a novel hierarchical architecture involving Ni-Mn LDHs@CuC2O4 nanosheets array on a copper mesh (CM). This study presents three significant contributions. We successfully synthesized the hierarchical Ni-Mn LDHs@CuC2O4 nanosheets array on CM. This architecture exhibits remarkable properties, including superior superhydrophilicity, underwater superoleophobicity, and self-cleaning capabilities. Lastly, we demonstrate the dual functionality of the Ni-Mn LDHs@CuC2O4 CM, serving as an effective medium for oil/water separation and as a promising supercapacitor electrode material. Our findings highlight the potential of this innovative design for practical applications in diverse fields.
Reviewer:
4. The authors should provide a clear schematic representation of the formation mechanism.
Response:
We appreciate for the reviewer’s good suggestion. Notably, Figure 1 has showed the schematic diagram of a two-step process for preparing Ni-Mn LDHs@CuC2O4 CM by chemical oxidation and hydrothermal deposition (please refer to page 3, Figure 1).
Reviewer:
5. What is the pH of the reaction solution? The pH of the solution normally varies from precursor to precursor. The authors must justify the selection of pH, temperature and time.
Response:
We appreciate the reviewer's helpful reminders. The reaction solution maintains a pH range of 9 to 10. Our investigation encompasses the impact of varying reaction times on the sample. With prolonged reaction times, an increasing number of nanosheets are observed to emerge from the membrane, contributing to the development of a more comprehensive structure. Notably, while numerous studies delve into LDHs, scant attention has been paid to the influence of reaction solution pH. Unfortunately, our research does not delve into this aspect.
Reviewer:
6. At what temperature did the authors perform the hydrothermal process?
Response:
We are grateful for the reviewer’s kindly reminder. We performed the hydrothermal process at 90 °C (please refer to page 12, line 320)
Reviewer:
7. The authors should include line representation in the XRD pattern.
Response:
We are thankful for the reviewer’s suggestion. The (0 0 3) planes confirming the successful synthesis of hydrotalcite has been marked in the Figure 4a (please refer to page 5, Figure 4a).
Reviewer:
8. There are many typo mistakes in the text, figure captions and some abbreviations are missing. Please revise carefully.
Response:
We are grateful for the reviewer’s insightful suggestion. We have carefully checked and corrected the errors in the text.
Reviewer:
9. The authors should discuss the effect of morphology and surface area on performance.
Response:
We appreciate the reviewer's insightful reminder. The hierarchical structure of Ni-Mn LDHs@CuC2O4 nanosheet arrays distinctly enhances the superhydrophilicity of the rough surface by effectively increasing the membrane's surface area, thereby augmenting surface roughness. (please refer to page 3, lines 107-114).
Reviewer:
10. What are the key factors affecting the efficiency?
Response:
We are grateful for the reviewer’s kindly reminder. The key factors include surface energy, roughness and surface area of the membrane.
Reviewer:
11. How do these results influence the previously reported results?
Response:
We are grateful for the reviewer’s kindly reminder. The high surface energy of copper mesh and the layered structure of hydrotalcite provide inspiration for our research.
Reviewer:
12, The authors should provide an EIS spectra of all samples.
Response:
We appreciate the reviewer's thoughtful reminder. Currently, the summer vacation is underway in China, which has posed challenges in successfully completing the EIS test. We kindly request the reviewers' understanding and support in this matter.
Reviewer:
13. Authors should provide CV and GCD curves of all samples.
Response:
Thank you for the reviewer’s thoughtful reminder, as the summer vacation is now underway in China. Regrettably, our attempts to successfully complete the CV and GCD tests have encountered challenges. We kindly request the understanding of the reviewers in this matter. Nevertheless, we have conducted a preliminary CV test on the CuC2O4 copper mesh sample. The outcome revealed a notable transformation as the samples turned black and subsequently decomposed within a 30-minute duration when immersed in a 6 M KOH electrolyte.
Reviewer:
14. All images are very poor resolution. Authors should produce high quality images.
Response:
We appreciate for the reviewer’s good suggestion. We have changed the fuzzy picture of the full text to improve the quality of the picture, including Figure 5, Figure 7, Figure 8 and Figure 9 (please refer to the latest Figure 5, Figure 7, Figure 8 and Figure 9).
Reviewer:
15. In the current state, there are more typographical errors and the language should be improved. Therefore, the authors are advised to recheck the whole manuscript for improving the language and structure carefully.
Response:
We are grateful for the reviewer’s insightful suggestion. We have carefully checked and corrected the errors in the text.
Round 2
Reviewer 2 Report
The authors have successfully answered most of the reviewer's concerns. The manuscript can now be accepted in present form.
English language looks fine. Minor checking and editing is required.
Reviewer 3 Report
The revised version can be acceptable in the present form.